# Cytosolic Release of Mitochondrial DNA and Associated cGAS Signaling Mediates Radiation-Induced Hematopoietic Injury of Mice

**DOI:** 10.3390/ijms24044020

**Published:** 2023-02-16

**Authors:** Hua Guan, Wen Zhang, Dafei Xie, Yuehua Nie, Shi Chen, Xiaoya Sun, Hongling Zhao, Xiaochang Liu, Hua Wang, Xin Huang, Chenjun Bai, Bo Huang, Pingkun Zhou, Shanshan Gao

**Affiliations:** 1Hengyang Medical School, University of South China, Hengyang 421001, China; 2Beijing Key Laboratory for Radiobiology, Department of Radiation Biology, Beijing Institute of Radiation Medicine, Beijing 100850, China; 3School of Public Health, University of South China, Hengyang 421001, China; 4Beijing Key Laboratory for Radiobiology, Department of Experimental Hematology and Biochemistry, Beijing Institute of Radiation Medicine, Beijing 100850, China

**Keywords:** mitochondrial DNA, bone marrow tissue, hematopoietic stem cells, radiation injury, non-target effect, cGAS signaling, inflammatory response, mice

## Abstract

Mitochondrion is an important organelle of eukaryotic cells and a critical target of ionizing radiation (IR) outside the nucleus. The biological significance and mechanism of the non-target effect originating from mitochondria have received much attention in the field of radiation biology and protection. In this study, we investigated the effect, role, and radioprotective significance of cytosolic mitochondrial DNA (mtDNA) and its associated cGAS signaling on hematopoietic injury induced by IR in vitro culture cells and in vivo total body irradiated mice in this study. The results demonstrated that γ-ray exposure increases the release of mtDNA into the cytosol to activate cGAS signaling pathway, and the voltage-dependent anion channel (VDAC) may contribute to IR-induced mtDNA release. VDAC1 inhibitor DIDS and cGAS synthetase inhibitor can alleviate bone marrow injury and ameliorate hematopoietic suppression induced by IR via protecting hematopoietic stem cells and adjusting subtype distribution of bone marrow cells, such as attenuating the increase of the F4/80^+^ macrophage proportion in bone marrow cells. The present study provides a new mechanistic explanation for the radiation non-target effect and an alternative technical strategy for the prevention and treatment of hematopoietic acute radiation syndrome.

## 1. Introduction

The bone marrow hematopoietic tissue generates all blood cell lineages from privileged niches in order to maintain a stable number of peripheral blood cells that are structurally and functionally connected to the immune system. It is well known that bone marrow is one of the most sensitive tissues to ionizing radiation (IR). The acute threshold dose is approximately 0.5 Gy (estimated dose for 1% incidence) for hematopoiesis depression. For mortality of the bone marrow syndrome without medical care, the threshold value is approximately 1 Gy of acutely absorbed dose [1]. Also, the accumulated data generated from humans exposed to radiation, either during radiation therapy or as a result of radiation accidents or nuclear explosions, suggests that the LD_50/60_ (i.e., 50% lethality dose at relatively high dose rates assessed 60 days after exposure) is approximately 3.3–4.5 Gy in the absence of medical management [1]. Hematopoietic acute radiation syndrome (H-ARS) results from excessive damage to hematopoietic progenitor cells (HPCs), leading to a heavy loss of blood cells. It is also characterized by altered serum cytokine levels and the loss of the hematopoietic stem cell (HSC) population. The increasing global risk of nuclear and radiological accidents or attacks has driven renewed research interest in developing medical countermeasures to potentially injurious exposures to acute irradiation. In addition, hematopoietic toxicity is also a common side effect of cancer radiotherapy. For example, the incidence of hematological toxicity is in the range of 30 to 70% for radiation therapy near the bony structures of pelvic tumors [2]. It is important to uncover the mechanism of radiation toxicity on hematopoiesis and develop medical countermeasures against H-ARS.

The targeted effect is a classical theory of radiobiology, and its central rule is that ionizing radiation (IR) directly acts on the target of organisms and leads to their death. The accumulated evidence support that nuclear genomic DNA is the main target of IR [3]. As a signal, DNA damage activates a series of biochemical reactions to respond to IR damage, triggering a variety of cellular reactions [4]. However, the discovery of non-target effects has challenged the nuclear genome-centered target effect, including bystander effects [5,6,7], genomic instability [3,8], and the abscopal effect in vivo [9]. The mechanism and weighting size of the non-target effect generated post-irradiation are not well addressed yet.

In fact, the early concept of a side effect stems from the increase of ROS in the mitochondrial pathway, which triggers complex intra- and intercellular signaling cascade reactions, leading to cell DNA damage, gene mutation, and other forms of radiation that mimic biological effects [10,11,12]. In addition, mitochondria are also the only place where DNA exists outside the nucleus. It is well known that the damage to nuclear DNA caused by IR includes multiple forms, such as double-strand breaks [13], resulting in base mismatches and gene fragment deletions [14,15], and this damage will also occur in mitochondrial DNA [16,17,18]. In the process of damage stress, mitochondria will choose to repair damaged DNA, degrade damaged proteins and lipids, or initiate a mitochondrial unfolded protein response to ensure the normal structure and function of mitochondria. When these rescue measures fail, mitochondria release mtDNA, which triggers an innate immune response [19]. The mitochondrial DNA is released into the cytoplasm, where it is recognized and activated by the cyclic GMP-AMP synthase (cGAS) [20,21]. More and more reports have confirmed that the activation of the cGAS-STING pathway shows great potential as an innate immunosensor in tumor cells and immune cells. Delivering corresponding agonists through appropriate drug delivery systems to enhance the immune system‘s ability to fight cancer can greatly promote the efficacy of cancer biotherapy [22,23]. In addition, the cytosolic DNA-cGAS/STING pathway can also treat cancer through radiation-induced non-targeted effects [21]. As a pattern recognition receptor (PRR), cGAS can detect various pathogen-associated molecular patterns (PAMPs) and danger-associated molecular patterns (DAMPs), and finally produce I-IFN [20,24]. Excessive production of I-IFN in tissues is harmful to cells and the body. There is already evidence that I-IFN can promote the reduction of bone marrow cells and the failure of hematopoietic stem cells (HSCs) [25,26,27]. If HSCs are injured or their self-renewal capacity is impaired, long-term or permanent damage to the hematopoietic system will occur, and it may lead to bone marrow failure and the death of organisms. Similarly, ionizing radiation can lead to bone marrow hematopoietic damage; the higher the dose is, the more serious the injury is. Without any medical treatment, hematopoietic injury caused by ionizing radiation may result in death [28].

In this study, in vitro and in vivo experiments were performed, and our results showed that ionizing radiation can cause elevated levels of cytosolic mtDNA in cultured mouse bone marrow FDC-P1 cells and mouse bone marrow cells. It was confirmed that ionizing radiation can induce the release of mtDNA from the mitochondrial matrix to the cytoplasm to activate the cGAS signaling pathway in the cytoplasm and finally participate in the hematopoietic tissue damage response. The radiation-induced bone marrow tissue injury can be partially mitigated by blocking the cytosolic release of mitochondrial DNA or inhibiting cGAS signaling.

## 2. Results

### 2.1. Increased Release of Mitochondrial DNA into Cytosol by γ-ray Irradiation

Mitochondrial DNA is reported to be more vulnerable to ionizing radiation damage than nuclear DNA [29,30,31] because it lacks histone protection [32]. On the other hand, when mitochondria are subjected to ionizing radiation, mitochondrial stress or other mechanisms may promote the release of mitochondrial DNA (mtDNA) fragments from mitochondria into the cytoplasm. Mice bone marrow FDC-P1 cells were irradiated with different doses (1 to 8 Gy) of ^60^Co γ-rays. Furthermore, the levels of mtDNA *ND*4, *ND*6, and *Cytb* gene fragments were detected in mitochondria and cytosol by real-time qPCR at 0.5 h after irradiation (Figure 1). Firstly, the mitochondria and cytosol components were isolated separately according to the method described previously [33]. The detection of mitochondrial proteins VDAC, COXIV, and the cytosolic component indicates that the extracted cytosol and mitochondria had no cross-contamination (Figure 1A). As shown in Figure 1B, the level of cytosolic mtDNA increased significantly after 2 Gy irradiation and reached its peak at 6 Gy. At the same time, the total mtDNA content in cells was detected and showed no change after irradiation in FDC-P1 (Figure 1C). The dynamics of mitochondrial DNA release into the cytosol were further analysed, with the level of cytosolic mtDNA increasing at 0.5 h after 6 Gy irradiation, reaching a peak at 12 h, and then decreasing (Figure 1D).

It was reported that mitochondrial membrane channel protein VDAC1 oligomerization can form pores in the mitochondrial outer membrane to promote the release of mtDNA [33,34]. Therefore, we pretreated FDC-P1 cells with DIDS, a VDAC1 inhibitor, for 0.5 h before irradiation to observe the effect of DIDS on mitochondrial DNA release. As shown in Figure 1E, the level of radiation-induced cytosolic mtDNA in DIDS-pretreated cells (DIDS+IR) was significantly reduced compared with that of irradiated cells alone.

The above results showed that ^60^Co γ-rays could promote the release of mtDNA from the mitochondrial matrix into the cytoplasm while DIDS could inhibit the release of mtDNA into cytoplasm from irradiated cells.

### 2.2. The Released Cytosolic mtDNA Involves in Regulation of FDC-P1 Cells’ Responses to γ-ray Irradiation

Cell cycle arrest is one of the main implications of eukaryotic cells’ responses to ionizing radiation. After 4 Gy irradiation, FDC-P1 cell growths were arrested at a certain phase at different time points post-irradiation (Figure 2A–D). A significantly increased ratio of G_2_ phase cells was induced at 6 –12 h after 4Gy irradiation (Figure 2B). As a results, the population of G_0_/G_1_ phase cells decreased at 12h (Figure 2C), and the S phase cells decreased post-irradiation at 6 and 12 h (Figure 2D). As mentioned above, DIDS can block the release of mitochondrial DNA into the cytosol. Obviously, pretreatment of DIDS at 0.5 h before 4 Gy irradiation can largely prevent the increase of G2 phase cells at 12 and 24 h post-irradiation.

Apoptosis is a critical endpoint of the cellular response triggered by IR-induced DNA damage signaling. After 2 Gy irradiation, a significantly increased ratio of apoptosis was induced in FDC-P1 cells. When DIDS was used to inhibit the cytosolic release of mtDNA, the yield of apoptosis induced by 2 Gy was significantly reduced. The above results indicate that the release of mtDNA into the cytoplasm after irradiation promoted the occurrence of cell cycle arrest and apoptosis, resulting in increased radiosensitivity of FDC-P1 cells.

### 2.3. Activation of the cGAS Signaling Pathway in Irradiated Cells by Cytosolic Mitochondrial DNA

As a cytoplasmic pattern recognition receptor, cGAS is activated by cytosolic mtDNAs. During cGAS signaling transduction, the phosphorylation of interferon factor 3 (IRF3) is mediated by the activated TBK1 kinase. In order to monitor the activation of cGAS signaling, the phosphorylation of its downstream targets TBK1 and IRF3 was detected. As shown in Figure 3A, the phosphorylated pTBK1 and pIRF3 were increased after irradiation, and these were attenuated by DIDS. The phosphorylated IRF3 in the irradiated cells was also suppressed by the cGAS inhibitor RU. 521 (Figure 3B). Type I IFN (IFN-I) is a group of inflammatory factors downstream of the cGAS signaling pathway. The phosphorylated IRF3 dimerizes and transfers to the nucleus to initiate the expression of IFN-I [20]. As shown in Figure 3C,D, the mRNA expression of IFN-α and INF-β significantly increased in FDC-P1 cells after being exposed to ^60^Co γ-rays. Furthermore, the radiation-increased expression of IFN-α and INF-β was inhibited by DIDS (Figure 3E,F) or the cGAS inhibitor RU. 521 (Figure 3G,H). These results indicate that the induction of INF-I inflammatory factors by ionizing radiation is, at least partially, related to the activation of the cGAS signaling pathway by the cytosolically released mtDNAs.

### 2.4. Cytosolic Release of Mitochondrial DNA Involved in Bone Marrow Tissue Injury of γ-ray Irradiated Mice

In vitro, we observed the release of mtDNA into the cytosol induced by ^60^Co γ-rays to activate cGAS signaling in FDC-P1 cells. We further confirmed a significantly increased level of cytosolic mtDNA in the bone marrow nucleated cells (BMNCs) of total body irradiated (TBI) C57BL/6N mice, which can be attenuated by DIDS (Figure 4A). The total mtDNA of ND4 and ND6 slightly increased in BMNCs. The results suggest that the cytosolic mtDNA-triggered signaling transduction may also be activated in the bone marrow tissue cells in vivo after total body irradiation.

On the basis of the above observations, the biological significance of radiation-induced mtDNA release to the cytoplasm was further evaluated by the changes in the hematological indices of mice. Expectedly, the peripheral white blood cell (WBC) count was significantly decreased in 6 Gy TBI irradiated mice. However, when compared with the irradiation alone group (6 Gy), the recovery of peripheral WBC count was clearly in the same dose irradiated mice combined with DIDS treatment (Figure 4C). Interestingly, in the irradiated mice with DIDS protection, the decrease in PLT count in peripheral blood was delayed, whereas the recovery was accelerated (Figure 4E). The decreasing of cell count appears to be delayed in the irradiated mice with DIDS prevention, but the recovery dynamic may remain unchanged when compared to the irradiated alone mice group (Figure 4F).

Bone marrow is one of the most sensitive tissues to ionizing radiation injuries, hematopoietic depression is a fatal tissue reaction. Therefore, we investigated the role of cytosolic mtDNA on TBI-induced bone marrow (BM) injury by histological analysis of mouse femur H&E staining at 48 h and 7 days after TBI (Figure 5). The femur pathological analysis of mice showed severe cell reduction and cell dissolution in irradiated mice (Figure 5A). Combined with DIDS protection, the number of cells in the femur of mice increased, and the degree of cell dissolution was largely remitted, indicating that blocking the cytosolic release of mtDNA can reduce the degree of bone marrow injury induced by ionizing radiation. Bone marrow nucleated cells (BMNCs) were counted at 48 h and 7 days after irradiation. As shown in Figure 5B,C, the radiation-caused decrease of BMNCs was ameliorated by DIDS.

To confirm that the activation of the cGAS signaling pathway by γ-ray irradiation is related to the released cytosolic mtDNA, mice were given the cGAS inhibitor RU. 521 10 minutes after irradiation, and pathological changes in bone marrow tissue were detected using H&E staining analysis 10 days later. Consistent with the previous observations (Figure 5A), the mouse femur bone marrow tissues remained quite few normal cells and displayed cell dissolution in the TBI mice, and which was largely ameliorated by RU. 521 (Figure 5D). The BMNCs were counted, and the results were consistent with the histopathological analysis (Figure 5E).

### 2.5. Radioprotection of Targeting Cytosolic mtDNA-cGAS Signaling Pathway Hematopoietic Cell Injury in Mice by Regulation

Hematopoietic stem cells (HSCs) are sensitive to ionizing radiation; HSCs’ death and exhaustion are the main reasons for hematopoietic failure of the BM induced by ionizing radiation (IR). It has been reported that IR-induced mitochondrial damage accelerates HSC injury [35]. On this basis, we studied the effect of blocking IR-induced mtDNA release on HSCs in mouse bone marrow 7 days after TBI. The level of HSCs in the bone marrow of irradiated mice was analyzed by flow cytometry with the relevant molecular markers (${\mathrm{Lin}}^{-}\mathrm{Sca}1^{+}c-{\mathrm{Kit}}^{+})$ (Figure 6). Irradiation of 6 Gy γ-ray led to a sharp decrease of HSCs in the bone marrow nucleated cells (BMNCs) and Lin(-)BMNCs (Figure 6A–F). Preventive treatment with DIDS significantly enhanced the levels of HSCs in BMNCs and Lin(-)BMNCs in the mouse bone marrow, 7 days after TBI (Figure 6B,C). The protective effects of the cGAS inhibitor RU.521 were further observed in HSCs in BMNCs and Lin (-) BMNCs in bone marrow tissue of TBI mice (Figure 6E,F).

The effects of cytolic mtDNA-related cGAS signaling on the subpopulation of bone marrow cells were further investigated in TBI mice. The changes in subtype distributions of bone marrow cells were measured on the 7th day after irradiation (Figure 7). Ly6C is a surface marker of monocytes related to differentiation; it is absent in erythroid cells and completely disappears after maturation into macrophages [36]. As shown in Figure 7, the proportion of Ly6C-positive monocytes in bone marrow cells is 0.93% in 6 Gy irradiation mice on the 7th day after irradiation, much lower than that of control mice (4.25%). When mice were pretreated with DIDS 1 h before irradiation, the proportions of Ly6C-positive cells increased from 0.93% to 2.59% (Figure 7B). F4/80 is a surface marker of macrophages, and the proportion of F4/80-positive cells in irradiated mice bone marrow cells is 2.38%, which is higher than of the proportion in control mice (0.71%). Pretreatment with DIDS could suppress the increase of F4/80 positive cells (Figure 7C).

The radioprotection of the cGAS inhibitor RU.521 on the subtype distribution of mouse bone marrow cells was further evaluated (Figure 7D–F). Mice were treated with RU. 521 10 min after 6Gy irradiation, and the subpopulations of bone marrow cells were also analyzed by flow cytometry 10 days after irradiation. Consistent with the effects of DIDS, RU.521 remarkedly ameliorated the alterations of Ly6C-positive (Figure 7E) and F4/80-positive cell (Figure 7F) subpopulations induced by γ-ray irradiation.

## 3. Discussion

This study demonstrated that the mouse hematopoietic FDC-P1 cells exposed to ^60^Co γ-rays caused an inflammatory response through the cGAS-STING signal transduction activated by the cytosolic mtDNA. Radiation exposure results in mtDNA leakage into the cytosolic matrix, which is the key mechanism for activating cGAS-STING signal transduction. We found that the voltage-dependent anion channel lying in the mitochondrial outer membrane contributes to radiation-induced mtDNA release, which can be blocked by the VDAC1 inhibitor DIDS. Mitochondrion is an important organelle of eukaryotic cells, and also acts as a target for various environmental factors [37,38,39,40,41]. Some damage stresses, such as H_2_O_2_, high glucose, LPS, and IR, were reported to increase mtDNA release into the cytosol [41,42,43,44,45,46]. One critical role of cytosolic mtDNA is to activate cGAS signal transduction and downstream inflammatory and immune responses [21]. Here, we confirmed that the increased cytosolic mtDNA plays a critical role in determining the sensitivity of cells to IR damage through activating the cGAS signal pathway (Figure 8). This mitochondria-originated effect is not dependent on the classical target effect of nuclear DNA damage [47], thus contributing novel mechanistic evidence supporting the non-target effect theory in the field of radiobiology.

There are a number of reports that, on the level of culture cells, demonstrate the increased cytosolic mtDNAs from nuclear-originated micronuclei or mtDNA fragment leakage from mitochondria, contribute to the cellular response to IR [48,49,50,51]. However, there are few such studies at the tissue or whole-body animal level. Hematopoietic tissue is very sensitive to ionizing radiation. Exposure to a certain dose of IR can cause hematopoietic damage, demonstrating bleeding, decreased blood cells, a reduced number of HSCs, bone marrow suppression, and bone marrow failure. In the current study, we found increased cytosolic release of mtDNA in bone marrow cells of TBI mice, which can exacerbate peripheral blood damage and bone marrow tissue and cell damage and delay the recovery of bone marrow tissue damage by activating the cGAS signaling pathway. These radiological effects can be ameliorated partially by DIDS-blocking mtDNA release and the cGAS inhibitor RU.521.

HSC injury is the main cause of hematopoietic exhaustion. The protection of HSCs from radiation damage should be a primary task. Previous studies have confirmed that the activation of cGAS can lead to the overexpression of type I IFN, in turn resulting in the depletion of long-term HSCs and hematopoietic injury, which indicates that cGAS-activating inflammatory factor production within HSCs participates in hematopoietic injury and can be protected by blocking cGAS synthase activity [25]. In addition, some inflammatory factors are involved in hematopoietic injury by indirectly regulating HSCs [52]. In this study, we found that blocking mtDNA release with DIDS or inhibiting cGAS activity with RU.521 can partially protect mouse bone marrow HSCs from radiation injury or hasten its recovery after 6 Gy of TBI. In the detection of various peripheral blood hemogram indexes, it was also found that the recovery of white blood cells began on the 7th day; WBC recovery is relatively early and much more efficient for the mice that received the protection of DIDS. Therefore, our study demonstrated that cytosolic mtDNA and its associated cGAS signal transduction pathway participate in radiation-induced hematopoietic injury. In addition, it was found that this cytosolic mtDNA-cGAS signal pathway also plays a role in the subtype population redistribution of bone marrow cells induced by IR, which includes its relation to a high ratio of the $F4/80^{+}$ (EGF-like module-containing mucin-like hormone receptor-like 1, EMR1) mature macrophage in the bone marrow during the process of recovery. Macrophage is a multifunctional subtype of bone marrow cells, and it is worth further studying the role and biological significance of cGAS signaling-associated $F4/80^{+}$ macrophage in IR-induced hematopoietic injury and recovery.

## 4. Materials and Methods

### 4.1. Cell Culture and Treatment

Mouse bone marrow FDC-P1 cells were purchased from the American Type Culture Collection (ATCC). FDC-P1 cells were cultured in RPMI1640 supplemented with 10% fetal bovine serum (FBS) and 0.5 ng/mL recombinant mouse IL-3 (Peprotech, No.213-13) at 37 °C with 5% CO_2_. DIDS sodium salt (MedChemExpress, No.67483-13-0) was reconstituted in sterile water at 500 mM of stock solution and used at 100 μM. DMSO was used to dissolve the powder of RU. 521 (MedChemExpress, No. HY-114180), and the concentration of mother liquor was 20 mM. The solution was frozen at −80 °C and diluted to 10 nM when used. At room temperature, cell cultures were irradiated with cobalt-60 γ-rays with the dose rate of 80.74 cGy/min [53].

### 4.2. Mice and Irradiation

The animal experiments were carried out on male C57BL/6N mice purchased from Vital River Laboratory Animal Co. (Beijing, China). Mice were aged 6–8 weeks when the experiments were performed, and they were housed and fed in standard laboratory conditions with a 12 h day-night cycle and controlled temperature (20–24 °C) and humidity (30–60%). Mice were irradiated with cobalt-60 γ-rays with a dose rate of 200 cGy/min at room temperature. All animal procedures and testing were conducted according to the China National Legislation of Laboratory Animal Welfare and Ethics and the local guidelines of the Laboratory Animals Center at the Academy of Military Medical Sciences (AMMS). Mice were treated humanely with regard to the alleviation of suffering. The study and research protocols were approved by the Institutional Committee for Animal Use and Care of AMMS.

### 4.3. Western Blot

The cells were lysed in M-PER^TM^ mammalian protein extraction reagent (Thermo Scientific, Waltham, MA, USA) [protease and phosphatase inhibitors (Selleck)], and the samples were incubated on ice for 20 min. Samples were centrifuged at 12,000 rpm for 15 min at 4 °C to yield the total protein for immunoblot analysis. Protein concentrations were measured with the BCA Protein Assay Kit (TIANGEN). Equal amounts of protein were separated by 10% SDS-PAGE and PVDF membranes. The membrane was blocked with 5% defatted milk for 2 h at room temperature, then incubated overnight at 4 °C with primary antibodies (Table 1). After the membrane was washed for 10 min each of three times in TBST, it was incubated for 2 h at room temperature with secondary antibodies. The blotted protein bands were visualized by SuperSignal™ West Pico PLUS Chemiluminescent Substrate (Thermo Scientific™) and exposed to the ImageQuant LAS 500.

### 4.4. Subcellular Fractionation and Quantification of Mitochondrial DNA

Subcellular fractionation and mitochondrial DNA quantification were adapted from West et al. and Kim et al. [33,34] as follows: 2 × 10^6^ FDC-P1 cells were lysed in 100 μL. Digitonin buffer [150 mM NaCl, 50 mM HEPES pH 7.4, and 25 µg/mL digitonin (Beyotime)] and incubated on a rotator at 4 °C for 10 min. Samples were centrifuged at 2000× *g* for 10 min at 4 °C. The supernatant was transferred to the fresh tube and centrifuged at 4 °C for 20 min at 20,000× *g*. Repeat three times. Between each centrifugation step, the supernatant was transferred to the fresh tube, and the cytosolic fractions containing cmtDNA were finally obtained. The qPCR was performed with 1:20 diluted cytosolic fractions for analysis of mitochondrial DNA using specific primers (Table 2). The remaining pellet from the first rotation was resuspended in ice-cold PBS to wash off the digitonin buffer. The samples were centrifuged at 2000× *g* for 5 min at 4 °C. The Blood/Cell/Tissue Genomic DNA Extraction Kit (TIANGEN) was used to extract total DNA, which contains total mtDNA, from the remaining pellet. The cmtDNA and total mtDNA are normalized to the total DNA in each sample particle.

### 4.5. RNA Extraction and Real-Time PCR

Total RNA in cells is extracted using the TRIzol reagent (Sigma, St. Louis, MO, USA) according to the manufacturer’s instructions. The quality of isolated RNA was controlled by an ultraviolet spectrophotometer (GE Healthcare GeneQuant 100, Chicago, IL, USA).An amount of 1 μg of RNA was used to synthesize cDNA using the PrimeScriptRT kit and gDNA Eraser (TaKaRa, Kusatsu, Japan). The real-time PCR analyses were performed on the Bio-Rad MyiQ™2 platform using the CFX96 TOUCH (USA) according to the manufacturer’s instructions. β-actin or GAPDH was used as a normalizer in the real-time PCR, and the relative expressions of evaluated genes were calculated by 2^–ΔΔCT^ method. Primer sequences are shown in Table 3.

### 4.6. Peripheral Blood Cell Counts

Peripheral blood was collected from the tail vein of mice to detect changes in the blood image after one month and analyzed on a hematology analyzer (Nihon Kohden, Tokyo, Japan). The cell counts included white blood cell (WBC) counts, red blood cell (RBC) counts, platelets (PLT), and lymphocytes (LYM).

### 4.7. Analysis of the Surface Markers of Mouse Bone Marrow Cells by Flow Cytometry

Femur samples were collected, bone marrow was extracted, single cells were collected, and they were centrifuged at 1500 rpm for 5 min at 4 °C, after which the supernatant was discarded. The pellet was lysed in 1 × RBC lysis buffer (Beyotime, Shanghai, China) for 15 min. The surface markers of each cell type were quantified by flow cytometry, and the number of cells used for analysis was 1 × 10^6/^tube.

Monoclonal antibodies FITC-Lin^-^, APC-c-kit, and PE-sca1 were used for the detection of LSK cells in bone marrow. Monoclonal antibodies against PE-Ly6C and FITC-F4/80 were used to detect changes in the number of monocytes and macrophages in mouse bone marrow; the flow antibody list is shown in the Table 4. The 1 × 10^6^ cells were washed twice with phosphate buffered saline (PBS), and resuspended in 100 µL PBS containing monoclonal antibodies and incubated at 4 °C for 30 min. The cells were then washed twice and resuspended in 200 µL PBS. A flow cytometer (Aria II, USA) was used for fluorescence analysis. Percentages of positive cells were evaluated based on fluorescence intensity.

### 4.8. Statistical Analysis

Unpaired numerical data were compared using unpaired *t*-tests (and nonparametric tests). The data were analyzed using the GraphPad Prism software (https://www.graphpad.com/scientific-software/prism, accessed on 5 April 2022).

## 5. Conclusions

In the current study, we demonstrated that γ-ray exposure increases the release of mtDNA into the cytosol, activating the cGAS signaling pathway, which plays a role in radiation-induced hematopoietic tissue injury in vitro and in vivo with TBI mice. Inhibiting the cytosolic mtDNA-cGAS signaling pathway can alleviate bone marrow tissue injury and ameliorate hematopoietic depression caused by IR, including radioprotection on HSCs and adjusting bone marrow cell subtype distribution. Our data provide a new mechanistic explanation for the radiation non-target effect and an alternative technical strategy for the prevention and treatment of hematopoietic acute radiation syndrome.

## Figures and Tables

**Figure 1 ijms-24-04020-f001:**
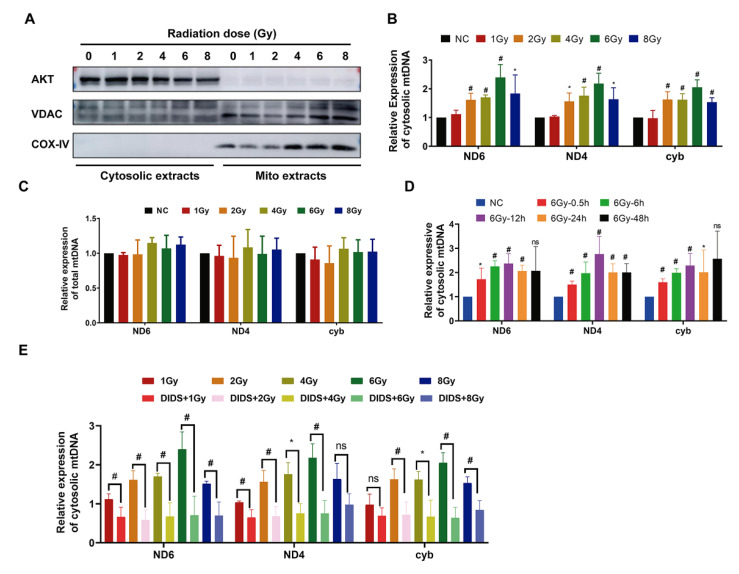
^60^Co γ-ray irradiation increased the release of mitochondrial DNA into cytosol. (**A**) Western blotting detects mitochondrial and cytosolic proteins to verify the fraction of different subcellular extracts; (**B**) the cytosolic mitochondrial DNA (ND4, ND6, and CytB gene fragments) detected by qPCR at 0.5 h after irradiation with different doses; (**C**) the total mitochondrial DNA (mtDNA) in the cells detected by qPCR at 0.5 h after irradiation with different doses; (**D**) the cytosolic mitochondrial DNA detected by qPCR at different times after 6 Gy irradiation with different doses; (**E**) the effect of VDAC1 inhibitor DIDS on release of mitochondrial DNA. Cells were pretreated with 100 nM DIDS at 0.5 h before irradiation, and the level of cytosolic mtDNA was detected by qPCR. The data are the mean ± SEM from three independent experiments: * *p* < 0.05, ^#^
*p* < 0.01.

**Figure 2 ijms-24-04020-f002:**
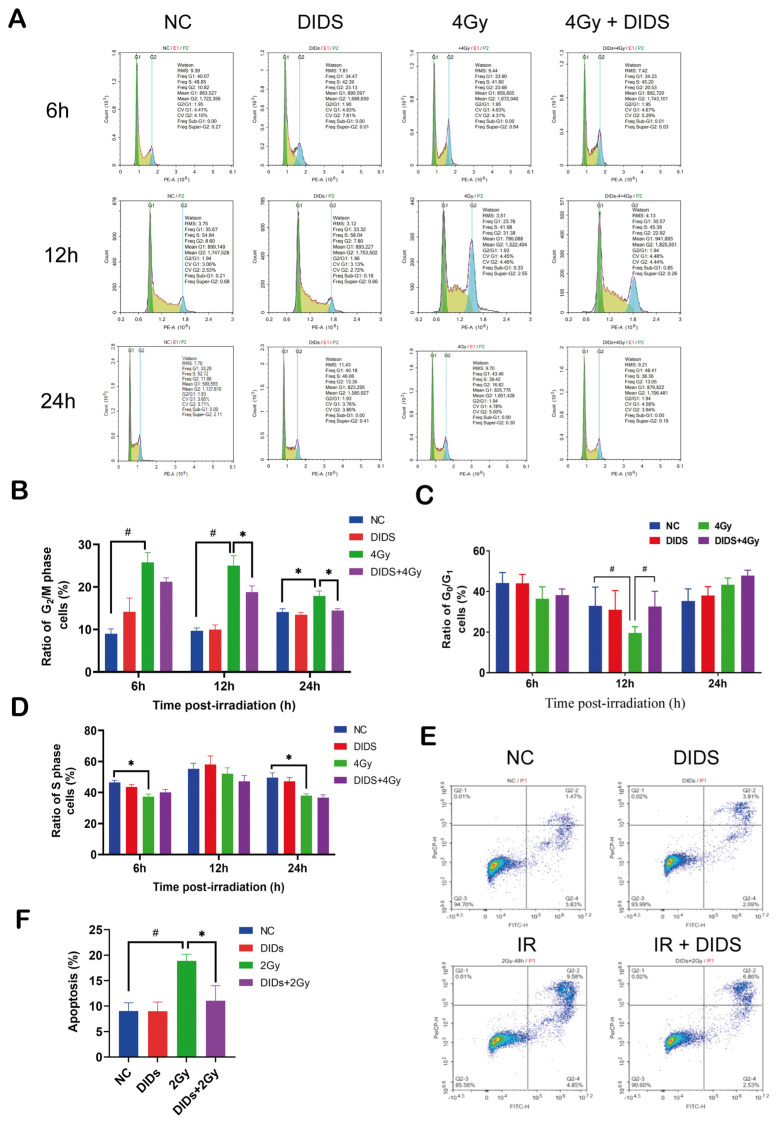
Cytosolic release of mitochondrial DNA participated in the regulation of FDC-P1 cells’ radiosensitivity. (**A**) Flow cytometric histograms showing the cell cycle arrest induced by γ-ray irradiation. FDC-P1 cells were irradiated with 4Gy Co^60^ γ-ray, with or without pretreatment of DIDS, at 0.5 h before irradiation. The cell cycle distributions at different time points after irradiation were detected; (**B**–**D**) ratio quantification of the cell cycle distribution; (**E**) flow cytometry images of apoptosis. FDC-P1 cells were pretreated with DIDS before the 2 Gy γ-ray irradiation, and apoptosis was detected 48 h after irradiation; (**F**) quantification of apoptosis ratio. The data are the mean ± SEM from three independent experiments: * *p* < 0.05, ^#^
*p* < 0.01.

**Figure 3 ijms-24-04020-f003:**
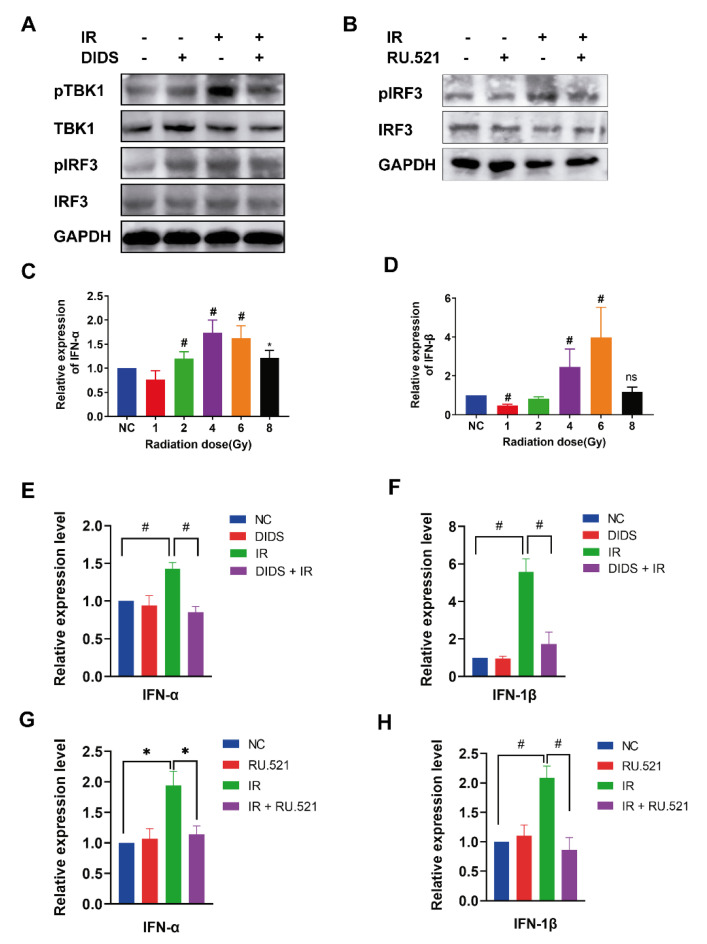
Radiation activates the cGAS signaling pathway by cytosolic mitochondrial DNA. (**A**) Radiation increased the phosphorylation of TBK1 and IRF3, which was detected by western blotting and suppressed by DIDS; (**B**) the cGAS inhibitor RU.521 attenuated radiation-induced phosphorylation of IRF3. Cells were treated with 10 nM RU.521 or 100 nM DIDS 1 h before irradiation, and the changes of the cGAS signaling pathway-related proteins were detected by Western blotting; (**C**,**D**) alterations of mRNA expression of IFN-α (**C**) and IFN-β (**D**) induced by γ-ray irradiation at different doses; (**E**–**H**) radiation-increased mRNA expression of IFN-α and IFN-β, and was suppressed by DIDS; (**E**,**F**) or RU.521 (**G**,**H**). The data are the mean ± SEM from three independent experiments: * *p* < 0.05, ^#^
*p* < 0.01.

**Figure 4 ijms-24-04020-f004:**
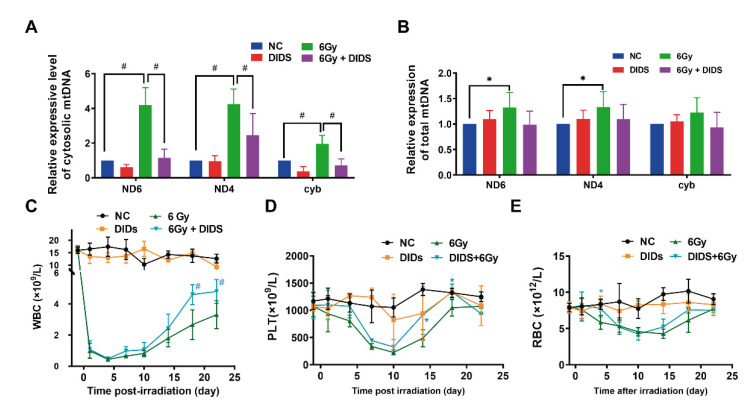
Increased level of cytosolic mtDNA in the BMNCc of 6Gy irradiated mice and the protective effects of DIDS on peripheral blood cells. (**A**) Cytosolic mtDNAs in bone marrow nucleated cells were detected by quantitative PCR. At 1 h before 6Gy irradiation, C57BL/6N mice were intraperitoneally injected with DIDS. The relative content of mtDNA in the cytosol of BMNCs was detected by qPCR at 6 h after irradiation; (**B**) total mtDNAs were detected in BMNCs of C57BL/6N mice; (**C**–**E**) the numbers of peripheral white blood cells; (**C**) platelets; (**D**) and red blood cells (**E**) were counted by a hematology analyzer. The data are the mean ± SEM from seven mice vs. the no-irradiated control (NC) group, * *p* < 0.05, ^#^
*p* < 0.01.

**Figure 5 ijms-24-04020-f005:**
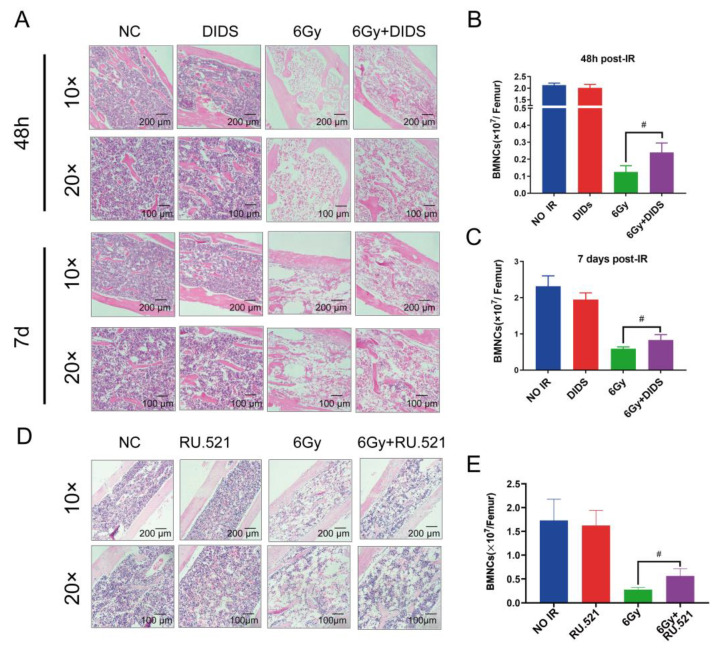
Activation of cGAS signaling by cytosolically released mtDNA involves in hematopoietic tissue injury of mice induced by γ-ray radiation. (**A**) Pathological analysis of mice femur tissue H&E staining. The mice were prophylactically administered DIDS by intraperitoneal injection 1 h before 6 Gy irradiation; H&E staining was performed to detect the pathological changes of femur tissue in mice 48 h and 7 d after irradiation; (**B**,**C**) the nucleated cells (BMNCs) in the bone marrow of mice were counted at 48 h (**B**) and 7 d (**C**) after TBI; (**D**) Pathological analysis of mice femur tissue (H&E staining). The mice were administered the cGAS inhibitor RU.521 by intraperitoneal injection at a dosage of 5 mg/kg at 10 min after 6 Gy irradiation. H&E staining was performed to detect the pathological changes of mouse femur tissue 10 days after irradiation. (**E**) Accordingly, the number of bone marrow nucleated cells in mice was counted 10 days after irradiation. The data are the mean ± SEM from six mice, ^#^
*p* < 0.01, vs. the 6 Gy alone group.

**Figure 6 ijms-24-04020-f006:**
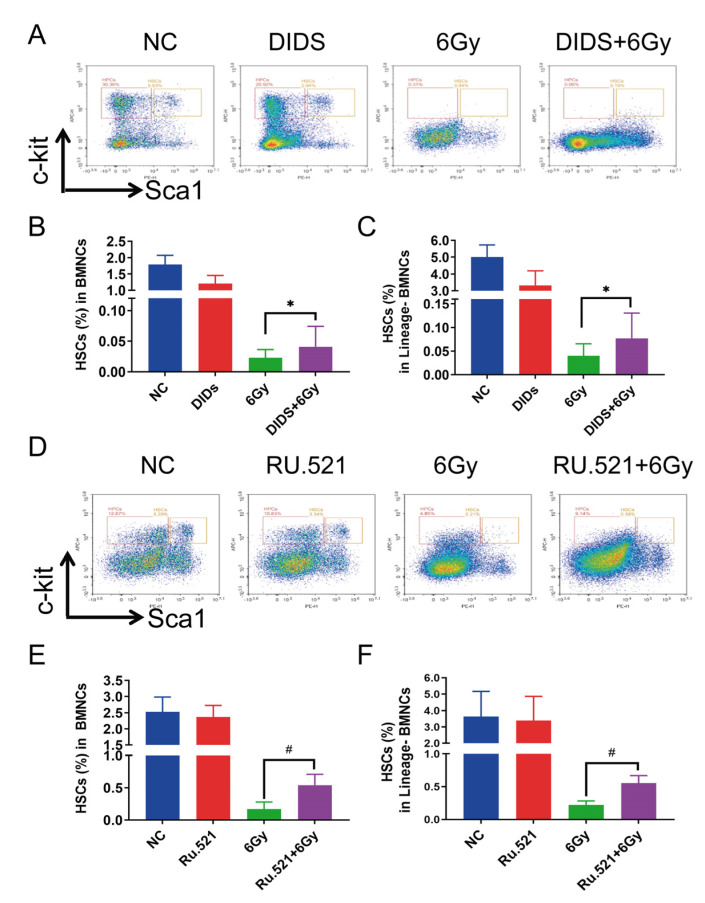
Protection of inhibiting cytosolic mtDNA-related cGAS signaling on hematopoietic stem cells in the bone marrow of TBI mice. (**A**) Flow cytometry analysis with relevant cell-specific molecular markers, to detect the proportion of HSCs in BMNC or Lin (-) BMNC cells in bone marrow tissue of 6Gy TBI mice and DIDS preventive effect 7 days after irradiation. For prevention with DIDS, mice were administered with DIDS by intraperitoneal injection 1 h before irradiation; (**B**,**C**) Percentage of HSCs in BMNC cells (**B**) and Lin (-) BMNCs (**C**) was quantitated; (**D**) Flow cytometry analysis with relevant cell-specific molecular markers to detect the proportion of HSCs in BMNC or Lin (-) BMNC cells in the bone marrow tissue of 6Gy TBI mice, and the prevention of RU.521 7 days after 6 Gy TBI. For prevention with RU.521, mice were administered with RU.521 by intraperitoneal injection 10 min after irradiation; (**E**,**F**) percentage of HSCs in BMNC cells (**E**) and Lin (-) BMNCs (**F**) was quantified. The data are the mean ± SEM from six mice, * *p* < 0.05, ^#^
*p* < 0.01, vs. the 6 Gy alone group.

**Figure 7 ijms-24-04020-f007:**
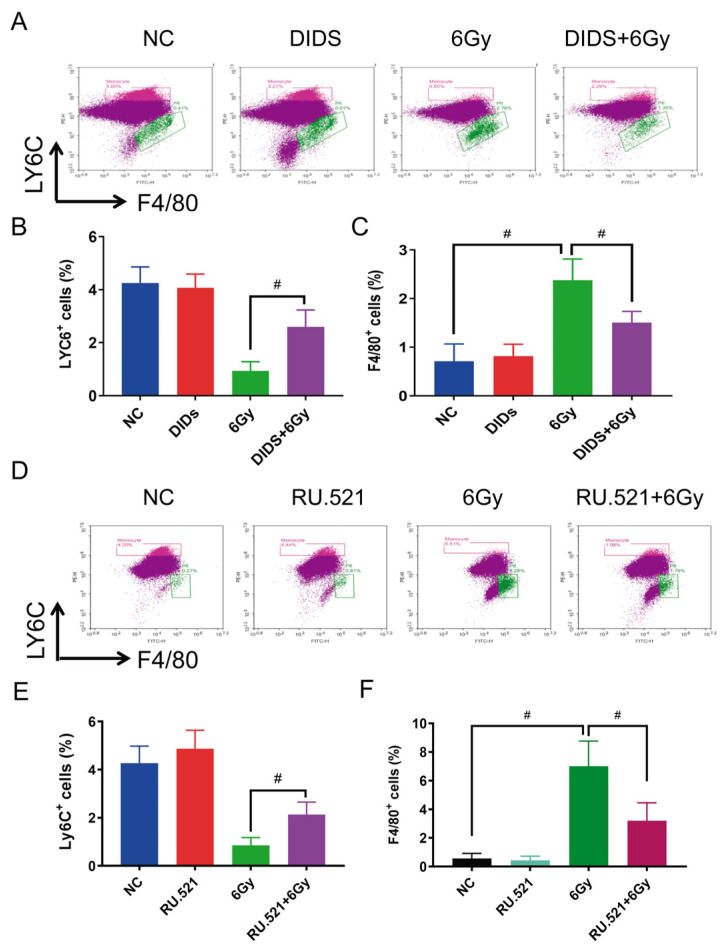
Protection of inhibiting cytosolic mtDNA-related cGAS signaling on monocyte/macrophage subtypes in bone marrow cells of TBI mice. (**A**) Visual histogram of flow cytometry analyzed with relevant molecular markers for monocytes (Ly6C) and macrophages (F4/80). The changes of the bone marrow nucleated cell subtypes were detected on the 7th day after 6Gy irradiation; (**B**,**C**) quantitative plots of Ly6C-positive monocyte percentage (**B**) and F4/80 positive macrophage percentage (**C**) and the protection of DIDS pretreated 1 h before irradiation. (**D**) A visual histogram of flow cytometry analyzed with relevant molecular markers for monocytes (Ly6C) and macrophages (F4/80). The changes of the bone marrow nucleated cell subtypes were detected on the 10th day after 6Gy irradiation; (**E**,**F**) quantitative plots of Ly6C-positive monocyte percentage (**E**) and F4/80-positive macrophage percentage (**F**) and the protection of RU.521 treated 10 min after 6Gy irradiation. The data are the mean ± SEM from six mice, ^#^
*p* < 0.01.

**Figure 8 ijms-24-04020-f008:**
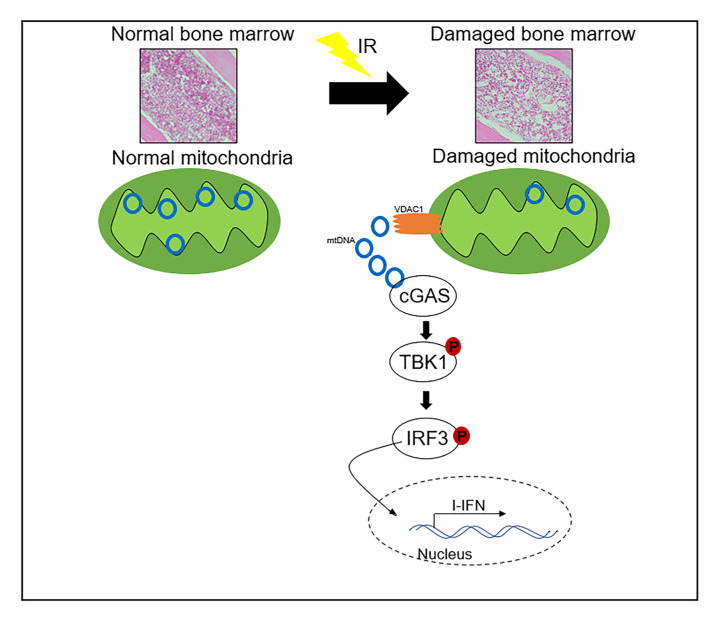
Ionizing radiation induced mtDNA release into the cytoplasm and activated the cGAS signaling pathway.

**Table 1 ijms-24-04020-t001:** Listing of antibodies used.

Name	Manufacturer	NO.	Dilution Ratio
anti-VDAC	CST	#4866	1:1000
anti-pIRF3(Ser396)	CST	#29037	1:1000
anti-IRF3	Santa	sc-33641	1:1000
anti-TBK1	Santa	Sc-52957	1:1000
Anti-pTBK1(ser172)	CST	#5483	1:1000
anti-COXIV	Proteintech	11242-1-AP	1:1000
anti-AKT	Proteintech	10176-2-AP	1:1000
anti-Lamin B1	Proteintech	12987-1-AP	1:1000

**Table 2 ijms-24-04020-t002:** List of PCR primer sequences.

Name	Primer Sequence (F)	Primer Sequence (R)
mmt-ND6	TTAGCATTAAAGCCTTCACC	CCAACAAACCCACTAACAAT
mmt-ND4	AACGGATCCACAGCCGTA	AGTCCTCGGGCCATGATT
mmt-cyb	GGCTACGTCCTTCCATGAGG	TCGGGTCAAGGTTGCTTTGT
GAPDH(mouse)	AGGAGCGAGACCCCACTAACA	AGGGGGGCTAAGCAGTTGGT

**Table 3 ijms-24-04020-t003:** List of primer sequences.

Name	Primer Sequence (F)	Primer Sequence (R)
Ifna (mouse)	ACCATCCCTGTCCTCCATGA	GAGGGTCTCATCCCAAGCAG
Ifnb (mouse)	TGACTATGGTCCAGGCACAG	TTGTTGAGAACCTCCTGGCT
Actin (mouse)	CACTGTCGAGTCGCGTC	GTCATCCATGGCGAACTGGT
GAPDH (mouse)	AGGAGCGAGACCCCACTAACA	AGGGGGGCTAAGCAGTTGGT

**Table 4 ijms-24-04020-t004:** Flow antibody inventory.

Name	Manufacturer	NO.
FITC anti-mouse Lineage Cocktail with	Biolegend	133302
APC anti-mouse CD117 (c-Kit) Antibody	Biolegend	105811
PE anti-mouse Ly-6A/E (Sca-1) Antibody	Biolegend	108107
PE anti-mouse Ly-6C	Biolegend	128007
F4/80 Monoclonal Antibody (BM8),FITC	eBioscience	11-4801-85

## Data Availability

The raw data supporting the data presented in this study are readily available on request from the corresponding author.

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
