# Peer review of "Cytosolic Release of Mitochondrial DNA and Associated cGAS Signaling Mediates Radiation-Induced Hematopoietic Injury of Mice"

_ijms, 2023, doi:10.3390/ijms24044020_

Round 1

Reviewer 1 Report

In this manuscript, Hua Guan et al. reported that γ-ray exposure leaded higher level of cytosolic mtDNA in cultured bone cells, and cGAS signaling pathway as well as voltage-dependent anion channel may involve in this process. Further, they found that VDAC1 and cGAS synthetase inhibitors alleviated bone marrow injury and ameliorate hematopoietic suppression induced by ionizing radiation. Though the study addresses an important area related to radiation injury and release of mitochondrial DNA, but the authors need to provide high quality figures, clarify their model and optimize the experimental design. Moreover, Further experiments are required and rationale is needed to delineate the model.

Below concerns should be address in a revised manuscript before acceptance for publication.

  1. The quality or resolution of all of figures.

The resolution of these figures is too low to get effective and detailed information. I can’t even distinguish the marks above the columns in Figure 1B, 1C, 1D and 1E. Is that an asterisk, a pound or a short dash? Similar situations occurred in all these figures, please provide high resolution figures.

  1. In Figure 1, the release of mitochondrial DNA into cytosol need to be reconfirmed by Southern blot (at least for Fig 1B&D). Because this is the most fundamental finding in current study, more solid data is required to support your conclusion.
  2. For Figure 1D, NC is not a good control. A more reasonable control is 6Gy-0h.
  3. A work model (maybe Figure 8) raised by your work would be very helpful for audience understanding your hypothesis.
  4. “*P < 0.05, **P < 0.01” is a more common usage for representing statistical significance.
  5. Lack of Data Availability statement.

Author Response

Dear Reviewer:

I quite appreciate your favorite consideration and the reviewer’s insightful comments. Now I have revised the ijms-2132860 exactly according to the reviewer’s comments, and found these comments are very helpful. I hope this revision can make my paper more acceptable. The revisions were addressed point by point below.

Reviewer 2 Report

This is an interesting article on a topic highly relevant to current trends on activation of sGas-STING pathway by radiation. Here the authors demonstrated in vitro and in vivo that cytosolic release of mtDNA activates cGAS signalling, causing an inflammatory response and hematopoietic damage. Additionally, the authors verified that reducing cytosolic releases of mtDNA by DISB-blocking and cGAS inhibitor Ru.521, significantly reduces the radiobiological effects in vitro and increases the hematopoietic recovery in vivo.

The manuscript is well written; however, I have some comments and suggestions:

1-    Introduction: If authors agree, a better description of the importance of cGAs-STING pathway has in cancer immunotherapy and how was been investigated for cancer treatment should be added to the manuscript.

2-    Figure 2A: subplots should have the same xx scale, having different scales make it very difficult to interpret this figure    

3-    Figure 3F and H: if I have interpreted this correctly, but plots corresponded to the same cell line treated with the same dose of radiation, only with different drugs (F with DIDS and H with RU.521), if this is correct, is somehow surprising that the relative expression of IFN-1B in the same cell line treated with the same dose of radiation  in the subplot F is around 5 and in the plot H is around 2, I would expect similar values as they are the same conditions. 

4-    Page 9 lines 229 to 232: Sentence starting with: “Consistent with previous observations, …” this sentence is confusing, and I would recommend the authors to rewrite it.

5-    Finally, would be of some interest to see how these concentrations of drugs (DIDS and RU.521) affect the normal function of mitochondria, would be good if authors could report some functional values with and without the drugs, such as mitochondria membrane potential.     

Author Response

(The authors gave the same response as above.)
